# QANet: Combining Local Convolution with Global Self-Attention for Reading Comprehension

**Adams Wei Yu**[1]*, **David Dohan**[2]†, **Minh-Thang Luong**[2]†
{weiyu}@cs.cmu.edu, {ddohan,thangluong}@google.com
[1]Carnegie Mellon University, [2]Google Brain

**Rui Zhao, Kai Chen, Mohammad Norouzi, Quoc V. Le**
Google Brain

## Abstract

Current end-to-end machine reading and question answering (Q&A) models are primarily based on recurrent neural networks (RNNs) with attention. Despite their success, these models are often slow for both training and inference due to the sequential nature of RNNs. We propose a new Q&A architecture called QANet, which does not require recurrent networks: Its encoder consists exclusively of convolution and self-attention, where convolution models local interactions and self-attention models global interactions. On the SQuAD dataset, our model is 3x to 13x faster in training and 4x to 9x faster in inference, while achieving equivalent accuracy to recurrent models. The speed-up gain allows us to train the model with much more data. We hence combine our model with data generated by back-translation from a neural machine translation model. On the SQuAD dataset, our single model, trained with augmented data, achieves 84.6 F1 score[1] on the test set, which is significantly better than the best published F1 score of 81.8.

## 1 Introduction

There is growing interest in the tasks of machine reading comprehension and automated question answering. Over the past few years, significant progress has been made with end-to-end models showing promising results on many challenging datasets. The most successful models generally employ two key ingredients: (1) a recurrent model to process sequential inputs, and (2) an attention component to cope with long term interactions. A successful combination of these two ingredients is the Bidirectional Attention Flow (BiDAF) model by Seo et al. (2016), which achieve strong results on the SQuAD dataset (Rajpurkar et al., 2016). A weakness of these models is that they are often slow for both training and inference due to their recurrent nature, especially for long texts. The expensive training not only leads to high turnaround time for experimentation and limits researchers from rapid iteration but also prevents the models from being used for larger dataset. Meanwhile the slow inference prevents the machine comprehension systems from being deployed in real-time applications.

In this paper, aiming to make the machine comprehension fast, we propose to remove the recurrent nature of these models. We instead exclusively use convolutions and self-attentions as the building blocks of encoders that separately encodes the query and context. Then we learn the interactions between context and question by standard attentions (Xiong et al., 2016; Seo et al., 2016; Bahdanau et al., 2015). The resulting representation is encoded again with our recurrency-free encoder before finally decoding to the probability of each position being the start or end of the answer span. We call this architecture QANet, which is shown in Figure 1.

---

*Work performed while Adams Wei Yu was with Google Brain.
†Equal contribution.
[1]While the major results presented here are those obtained in Oct 2017, our latest scores (as of Apr 23, 2018) on SQuAD leaderboard is EM/F1=82.2/88.6 for single model and EM/F1=83.9/89.7 for ensemble, both ranking No.1. Notably, the EM of our ensemble is better than the human performance (82.3).

The key motivation behind the design of our model is the following: convolution captures the local structure of the text, while the self-attention learns the global interaction between each pair of words. The additional context-query attention is a standard module to construct the query-aware context vector for each position in the context paragraph, which is used in the subsequent modeling layers. The feed-forward nature of our architecture speeds up the model significantly. In our experiments on the SQuAD dataset, our model is 3x to 13x faster in training and 4x to 9x faster in inference. As a simple comparison, our model can achieve the same accuracy (77.0 F1 score) as BiDAF model (Seo et al., 2016) within 3 hours training that otherwise should have taken 15 hours. The speed-up gain also allows us to train the model with more iterations to achieve better results than competitive models. For instance, if we allow our model to train for 18 hours, it achieves an F1 score of 82.7 on the dev set, which is much better than (Seo et al., 2016), and is on par with best published results.

As our model is fast, we can train it with much more data than other models. To further improve the model, we propose a complementary data augmentation technique to enhance the training data. This technique paraphrases the examples by translating the original sentences from English to another language and then back to English, which not only enhances the number of training instances but also diversifies the phrasing.

On the SQuAD dataset, QANet trained with the augmented data achieves 84.6 F1 score on the test set, which is significantly better than the best published result of 81.8 by Hu et al. (2017).[2] We also conduct ablation test to justify the usefulness of each component of our model. In summary, the contribution of this paper are as follows:

- We propose an efficient reading comprehension model that exclusively built upon convolutions and self-attentions. To the best of our knowledge, we are the first to do so. This combination maintains good accuracy, while achieving up to 13x speedup in training and 9x per training iteration, compared to the RNN counterparts. The speedup gain makes our model the most promising candidate for scaling up to larger datasets.

- To improve our result on SQuAD, we propose a novel data augmentation technique to enrich the training data by paraphrasing. It allows the model to achieve higher accuracy that is better than the state-of-the-art.

## 2 THE MODEL

In this section, we first formulate the reading comprehension problem and then describe the proposed model QANet: it is a feedforward model that consists of only convolutions and self-attention, a combination that is empirically effective, and is also a novel contribution of our work.

### 2.1 PROBLEM FORMULATION

The reading comprehension task considered in this paper, is defined as follows. Given a context paragraph with $n$ words $C = \{c_1, c_2, ..., c_n\}$ and the query sentence with $m$ words $Q = \{q_1, q_2, ..., q_m\}$, output a span $S = \{c_i, c_{i+1}, ..., c_{i+j}\}$ from the original paragraph $C$. In the following, we will use $x$ to denote both the original word and its embedded vector, for any $x \in C, Q$.

### 2.2 MODEL OVERVIEW

The high level structure of our model is similar to most existing models that contain five major components: an embedding layer, an embedding encoder layer, a context-query attention layer, a model encoder layer and an output layer, as shown in Figure 1. These are the standard building blocks for most, if not all, existing reading comprehension models. However, the major differences between our approach and other methods are as follow: For both the embedding and modeling encoders, we only use convolutional and self-attention mechanism, discarding RNNs, which are used by most of the existing reading comprehension models. As a result, our model is much faster, as it can process the input tokens in parallel. Note that even though self-attention has already been

---

[2] After our first submission of the draft, there are other unpublished results either on the leaderboard or arxiv. For example, the current (as of Dec 19, 2017) best documented model, SAN Liu et al. (2017b), achieves 84.4 F1 score which is on par with our method.

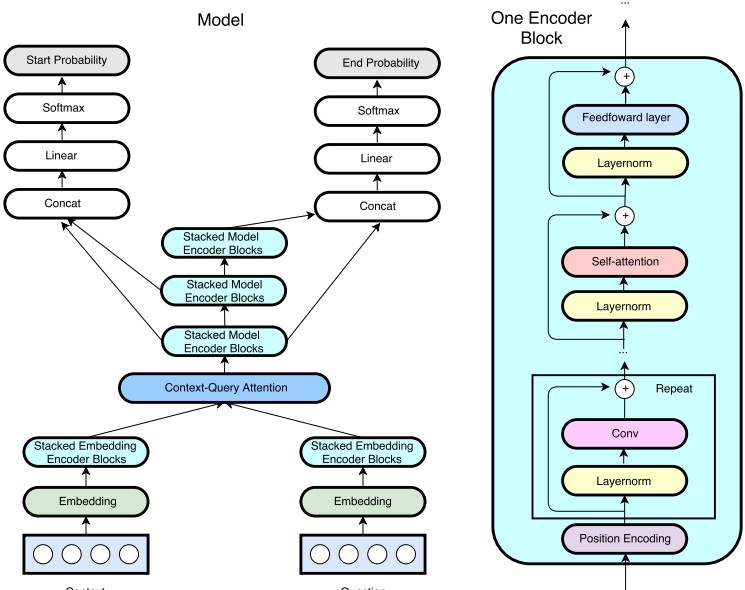

Figure 1: An overview of the QANet architecture (left) which has several Encoder Blocks. We use the same Encoder Block (right) throughout the model, only varying the number of convolutional layers for each block. We use layernorm and residual connection between every layer in the Encoder Block. We also share weights of the context and question encoder, and of the three output encoders. A positional encoding is added to the input at the beginning of each encoder layer consisting of $sin$ and $cos$ functions at varying wavelengths, as defined in (Vaswani et al., 2017a). Each sub-layer after the positional encoding (one of convolution, self-attention, or feed-forward-net) inside the encoder structure is wrapped inside a residual block.

used extensively in Vaswani et al. (2017a), the combination of convolutions and self-attention is novel, and is significantly better than self-attention alone and gives 2.7 F1 gain in our experiments. The use of convolutions also allows us to take advantage of common regularization methods in ConvNets such as stochastic depth (layer dropout) (Huang et al., 2016), which gives an additional gain of 0.2 F1 in our experiments.

In detail, our model consists of the following five layers:

**1. Input Embedding Layer.** We adopt the standard techniques to obtain the embedding of each word $w$ by concatenating its word embedding and character embedding. The word embedding is fixed during training and initialized from the $p_1 = 300$ dimensional pre-trained GloVe (Pennington et al., 2014) word vectors, which are fixed during training. All the out-of-vocabulary words are mapped to an `<UNK>` token, whose embedding is trainable with random initialization. The character embedding is obtained as follows: Each character is represented as a trainable vector of dimension $p_2 = 200$, meaning each word can be viewed as the concatenation of the embedding vectors for each of its characters. The length of each word is either truncated or padded to 16. We take maximum value of each row of this matrix to get a fixed-size vector representation of each word. Finally, the output of a given word $x$ from this layer is the concatenation $[x_w; x_c] \in \mathbf{R}^{p_1+p_2}$, where $x_w$ and $x_c$ are the word embedding and the convolution output of character embedding of $x$ respectively. Following Seo et al. (2016), we also adopt a two-layer highway network (Srivastava et al., 2015) on top of this representation. For simplicity, we also use $x$ to denote the output of this layer.

**2. Embedding Encoder Layer.** The encoder layer is a stack of the following basic building block: [convolution-layer × # + self-attention-layer + feed-forward-layer], as illustrated in the upper right of Figure 1. We use depthwise separable convolutions (Chollet, 2016) (Kaiser et al., 2017) rather than traditional ones, as we observe that it is memory efficient and has better generalization. The kernel size is 7, the number of filters is $d = 128$ and the number of conv layers within a block is

4. For the self-attention-layer, we adopt the multi-head attention mechanism defined in (Vaswani et al., 2017a) which, for each position in the input, called the query, computes a weighted sum of all positions, or keys, in the input based on the similarity between the query and key as measured by the dot product. The number of heads is 8 throughout all the layers. Each of these basic operations (conv/self-attention/ffn) is placed inside a *residual block*, shown lower-right in Figure 1. For an input $x$ and a given operation $f$, the output is $f(layernorm(x)) + x$, meaning there is a full identity path from the input to output of each block, where layernorm indicates layer-normalization proposed in (Ba et al., 2016). The total number of encoder blocks is 1. Note that the input of this layer is a vector of dimension $p_1 + p_2 = 500$ for each individual word, which is immediately mapped to $d = 128$ by a one-dimensional convolution. The output of this layer is a also of dimension $d = 128$.

**3. Context-Query Attention Layer.** This module is standard in almost every previous reading comprehension models such as Weissenborn et al. (2017) and Chen et al. (2017). We use $C$ and $Q$ to denote the encoded context and query. The context-to-query attention is constructed as follows: We first computer the similarities between each pair of context and query words, rendering a similarity matrix $S \in \mathbf{R}^{n \times m}$. We then normalize each row of $S$ by applying the softmax function, getting a matrix $\overline{S}$. Then the context-to-query attention is computed as $A = \overline{S} \cdot Q^T \in \mathbf{R}^{n \times d}$. The similarity function used here is the trilinear function (Seo et al., 2016):

$$f(q, c) = W_0[q, c, q \odot c],$$

where $\odot$ is the element-wise multiplication and $W_0$ is a trainable variable.

Most high performing models additionally use some form of query-to-context attention, such as BiDaF (Seo et al., 2016) and DCN (Xiong et al., 2016). Empirically, we find that, the DCN attention can provide a little benefit over simply applying context-to-query attention, so we adopt this strategy. More concretely, we compute the column normalized matrix $\overline{\overline{S}}$ of $S$ by softmax function, and the query-to-context attention is $B = \overline{S} \cdot \overline{\overline{S}}^T \cdot C^T$.

**4. Model Encoder Layer.** Similar to Seo et al. (2016), the input of this layer at each position is $[c, a, c \odot a, c \odot b]$, where $a$ and $b$ are respectively a row of attention matrix $A$ and $B$. The layer parameters are the same as the Embedding Encoder Layer except that convolution layer number is 2 within a block and the total number of blocks are 7. We share weights between each of the 3 repetitions of the model encoder.

**5. Output layer.** This layer is task-specific. Each example in SQuAD is labeled with a span in the context containing the answer. We adopt the strategy of Seo et al. (2016) to predict the probability of each position in the context being the start or end of an answer span. More specifically, the probabilities of the starting and ending position are modeled as

$$p^1 = softmax(W_1[M_0; M_1]), \ p^2 = softmax(W_2[M_0; M_2]),$$

where $W_1$ and $W_2$ are two trainable variables and $M_0, M_1, M_2$ are respectively the outputs of the three model encoders, from bottom to top. The score of a span is the product of its start position and end position probabilities. Finally, the objective function is defined as the negative sum of the log probabilities of the predicted distributions indexed by true start and end indices, averaged over all the training examples:

$$L(\theta) = -\frac{1}{N} \sum_i^N \left[ \log(p^1_{y_i^1}) + \log(p^2_{y_i^2}) \right],$$

where $y_i^1$ and $y_i^2$ are respectively the groundtruth starting and ending position of example $i$, and $\theta$ contains all the trainable variables. The proposed model can be customized to other comprehension tasks, e.g. selecting from the candidate answers, by changing the output layers accordingly.

**Inference**. At inference time, the predicted span $(s, e)$ is chosen such that $p^1_s p^2_e$ is maximized and $s \le e$. Standard dynamic programming can obtain the result with linear time.

## 3 DATA AUGMENTATION BY BACKTRANSLATION

Since our model is fast, we can train it with much more data. We therefore combine our model with a simple data augmentation technique to enrich the training data. The idea is to use two trans-

lation models, one translation model from English to French (or any other language) and another translation model from French to English, to obtain paraphrases of texts. This approach helps automatically increase the amount of training data for broadly any language-based tasks including the reading comprehension task that we are interested in. With more data, we expect to better regularize our models. The augmentation process is illustrated in Figure 2 with French as a pivotal language.

In this work, we consider attention-based neural machine translation (NMT) models Bahdanau et al. (2015); Luong et al. (2015), which have demonstrated excellent translation quality Wu et al. (2016), as the core models of our data augmentation pipeline. Specifically, we utilize the publicly available codebase[3] provided by Luong et al. (2017), which replicates the Google's NMT (GNMT) systems Wu et al. (2016). We train 4-layer GNMT models on the public WMT data for both English-French[4] (36M sentence pairs) and English-German[5] (4.5M sentence pairs). All data have been tokenized and split into subword units as described in Luong et al. (2017). All models share the same hyperparameters[6] and are trained with different numbers of steps, 2M for English-French and 340K for English-German. Our English-French systems achieve 36.7 BLEU on newstest2014 for translating into French and 35.9 BLEU for the reverse direction. For English-German and on newstest2014, we obtain 27.6 BLEU for translating into German and 29.9 BLEU for the reverse direction.

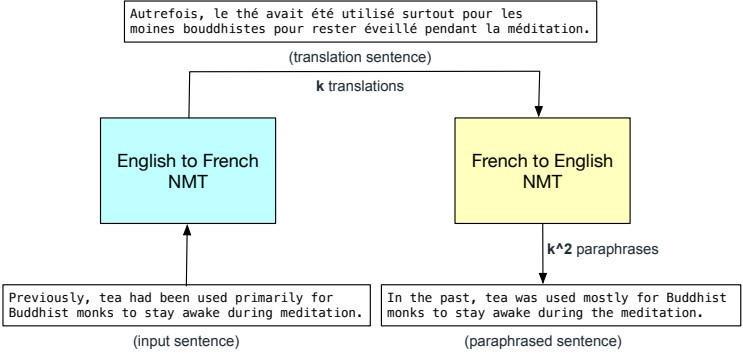

Figure 2: An illustration of the data augmentation process with French as a pivotal language. **k** is the beam width, which is the number of translations generated by the NMT system.

Our paraphrase process works as follows, supposedly with French as a pivotal language. First, we feed an input sequence into the beam decoder of an English-to-French model to obtain $k$ French translations. Each of the French translation is then passed through the beam decoder of a reversed translation model to obtain a total of $k^2$ paraphrases of the input sequence.

**Relation to existing Works.** While the concept of backtranslation has been introduced before, it is often used to improve either the same translation task Sennrich et al. (2016) or instrinsic paraphrase evaluations Wieting et al. (2017); Mallinson et al. (2017). Our approach is a novel application of backtranslation to enrich training data for down-stream tasks, in this case, the question answering (QA) task. It is worth to note that (Dong et al., 2017) use paraphrasing techniques to improve QA; however, they only paraphrase questions and did not focus on the data augmentation aspect as we do in this paper.

**Handling SQuAD Documents and Answers.** We now discuss our specific procedure for the SQuAD dataset, which is essential for best performance gains. Remember that, each training example of SQuAD is a triple of $(d, q, a)$ in which document $d$ is a multi-sentence paragraph that has the answer $a$. When paraphrasing, we keep the question $q$ unchanged (to avoid accidentally changing its meaning) and generate new triples of $(d', q, a')$ such that the new document $d'$ has the new answer

---

[3] https://github.com/tensorflow/nmt
[4] http://www.statmt.org/wmt14/
[5] http://www.statmt.org/wmt16/
[6] https://github.com/tensorflow/nmt/blob/master/nmt/standard_hparams/wmt16_gnmt_4_layer.json

$a'$ in it. The procedure happens in two steps: (i) *document paraphrasing* – paraphrase $d$ into $d'$ and (b) *answer extraction* – extract $a'$ from $d'$ that closely matches $a$.

For the document paraphrasing step, we first split paragraphs into sentences and paraphrase them independently. We use $k = 5$, so each sentence has 25 paraphrase choices. A new document $d'$ is formed by simply replacing each sentence in $d$ with a randomly-selected paraphrase. An obvious issue with this naïve approach is that the original answer $a$ might no longer be present in $d'$.

The answer extraction addresses the aforementioned issue. Let $s$ be the original sentence that contains the original answer $a$ and $s'$ be its paraphrase. We identify the newly-paraphrased answer with simple heuristics as follows. Character-level 2-gram scores are computed between each word in $s'$ and the start / end words of $a$ to find start and end positions of possible answers in $s'$. Among all candidate paraphrased answer, the one with the highest character 2-gram score with respect to $a$ is selected as the new answer $a'$. Table 1 shows an example of the new answer found by this process.[7]

| | Sentence that contains an answer | Answer |
|---|---|---|
| Original | All of the departments in the College of Science offer PhD programs, except for the Department of Pre-Professional Studies. | Department of Pre-Professional Studies |
| Paraphrase | All departments in the College of Science offer PHD programs with the exception of the Department of Preparatory Studies. | Department of Preparatory Studies |

Table 1: Comparison between answers in original sentence and paraphrased sentence.

The quality and diversity of paraphrases are essential to the data augmentation method. It is still possible to improve the quality and diversity of this method. The quality can be improved by using better translation models. For example, we find paraphrases significantly longer than our models' maximum training sequence length tend to be cut off in the middle. The diversity can be improved by both sampling during the beam search decoding and paraphrasing questions and answers in the dataset as well. In addition, we can combine this method with other data augmentation methods, such as, the type swap method (Raiman & Miller, 2017), to acquire more diversity in paraphrases.

In our experiments, we observe that the proposed data augmentation can bring non-trivial improvement in terms of accuracy. We believe this technique is also applicable to other supervised natural language processing tasks, especially when the training data is insufficient.

## 4 EXPERIMENTS

In this section, we conduct experiments to study the performance of our model and the data augmentation technique. We will primarily benchmark our model on the SQuAD dataset (Rajpurkar et al., 2016), considered to be one of the most competitive datasets in Q&A. We also conduct similar studies on TriviaQA (Joshi et al., 2017), another Q&A dataset, to show that the effectiveness and efficiency of our model are general.

### 4.1 EXPERIMENTS ON SQUAD

#### 4.1.1 DATASET AND EXPERIMENTAL SETTINGS

**Dataset.** We consider the Stanford Question Answering Dataset (SQuAD) (Rajpurkar et al., 2016) for machine reading comprehension.[8] SQuAD contains 107.7K query-answer pairs, with 87.5K for training, 10.1K for validation, and another 10.1K for testing. The typical length of the paragraphs is around 250 while the question is of 10 tokens although there are exceptionally long cases. Only the training and validation data are publicly available, while the test data is hidden that one has to submit the code to a Codalab and work with the authors of (Rajpurkar et al., 2016) to retrieve the final test

---

[7]We also define a minimum threshold for elimination. If there is no answer with 2-gram score higher than the threshold, we remove the paraphrase $s'$ from our sampling process. If all paraphrases of a sentence are eliminated, no sampling will be performed for that sentence.

[8]SQuAD leaderboard: `https://rajpurkar.github.io/SQuAD-explorer/`

score. In our experiments, we report the test set result of our best single model.[9] For further analysis, we only report the performance on the validation set, as we do not want to probe the unseen test set by frequent submissions. According to the observations from our experiments and previous works, such as (Seo et al., 2016; Xiong et al., 2016; Wang et al., 2017; Chen et al., 2017), the validation score is well correlated with the test score.

**Data Preprocessing.** We use the NLTK tokenizer to preprocess the data.[10] The maximum context length is set to 400 and any paragraph longer than that would be discarded. During training, we batch the examples by length and dynamically pad the short sentences with special symbol <PAD>. The maximum answer length is set to 30. We use the pretrained 300-D word vectors GLoVe (Pennington et al., 2014), and all the out-of-vocabulary words are replace with <UNK>, whose embedding is updated during training. Each character embedding is randomly initialized as a 200-D vector, which is updated in training as well. We generate two additional augmented datasets obtained from Section 3, which contain 140K and 240K examples and are denoted as "data augmentation × 2" and "data augmentation × 3" respectively, including the original data.

**Training details.** We employ two types of standard regularizations. First, we use L2 weight decay on all the trainable variables, with parameter $\lambda = 3 \times 10^{-7}$. We additionally use dropout on word, character embeddings and between layers, where the word and character dropout rates are 0.1 and 0.05 respectively, and the dropout rate between every two layers is 0.1. We also adopt the stochastic depth method (layer dropout) (Huang et al., 2016) within each embedding or model encoder layer, where sublayer $l$ has survival probability $p_l = 1 - \frac{l}{L}(1 - p_L)$ where $L$ is the last layer and $p_L = 0.9$.

The hidden size and the convolution filter number are all 128, the batch size is 32, training steps are 150K for original data, 250K for "data augmentation × 2", and 340K for "data augmentation × 3". The numbers of convolution layers in the embedding and modeling encoder are 4 and 2, kernel sizes are 7 and 5, and the block numbers for the encoders are 1 and 7, respectively.

We use the ADAM optimizer (Kingma & Ba, 2014) with $\beta_1 = 0.8, \beta_2 = 0.999, \epsilon = 10^{-7}$. We use a learning rate warm-up scheme with an inverse exponential increase from 0.0 to 0.001 in the first 1000 steps, and then maintain a constant learning rate for the remainder of training. Exponential moving average is applied on all trainable variables with a decay rate 0.9999.

Finally, we implement our model in Python using Tensorflow (Abadi et al., 2016) and carry out our experiments on an NVIDIA p100 GPU.[11]

### 4.1.2 RESULTS

**Accuracy.** The F1 and Exact Match (EM) are two evaluation metrics of accuracy for the model performance. F1 measures the portion of overlap tokens between the predicted answer and groundtruth, while exact match score is 1 if the prediction is exactly the same as groundtruth or 0 otherwise. We show the results in comparison with other methods in Table 2. To make a fair and thorough comparison, we both report both the published results in their latest papers/preprints and the updated but not documented results on the leaderboard. We deem the latter as the unpublished results. As can be seen from the table, the accuracy (EM/F1) performance of our model is on par with the state-of-the-art models. In particular, our model trained on the original dataset outperforms all the documented results in the literature, in terms of both EM and F1 scores (see second column of Table 2). When trained with the augmented data with proper sampling scheme, our model can get significant gain 1.5/1.1 on EM/F1. Finally, our result on the official test set is 76.2/84.6, which significantly outperforms the best documented result 73.2/81.8.

**Speedup over RNNs.** To measure the speedup of our model against the RNN models, we also test the corresponding model architecture with each encoder block replaced with a stack of bidirectional

---

[9]On the leaderboard of SQuAD, there are many strong candidates in the "ensemble" category with high EM/F1 scores. Although it is possible to improve the results of our model using ensembles, we focus on the "single model" category and compare against other models with the same category.

[10]NLTK implementation: http://www.nltk.org/

[11]TensorFlow implementation: https://www.tensorflow.org/

[12]The scores are collected from the latest version of the documented related work on Oct 27, 2017.

[13]The scores are collected from the leaderboard on Oct 27, 2017.

| | Published[12] | LeaderBoard[13] |
|---|---|---|
| Single Model | EM / F1 | EM / F1 |
| LR Baseline (Rajpurkar et al., 2016) | 40.4 / 51.0 | 40.4 / 51.0 |
| Dynamic Chunk Reader (Yu et al., 2016) | 62.5 / 71.0 | 62.5 / 71.0 |
| Match-LSTM with Ans-Ptr (Wang & Jiang, 2016) | 64.7 / 73.7 | 64.7 / 73.7 |
| Multi-Perspective Matching (Wang et al., 2016) | 65.5 / 75.1 | 70.4 / 78.8 |
| Dynamic Coattention Networks (Xiong et al., 2016) | 66.2 / 75.9 | 66.2 / 75.9 |
| FastQA (Weissenborn et al., 2017) | 68.4 / 77.1 | 68.4 / 77.1 |
| BiDAF (Seo et al., 2016) | 68.0 / 77.3 | 68.0 / 77.3 |
| SEDT (Liu et al., 2017a) | 68.1 / 77.5 | 68.5 / 78.0 |
| RaSoR (Lee et al., 2016) | 70.8 / 78.7 | 69.6 / 77.7 |
| FastQAExt (Weissenborn et al., 2017) | 70.8 / 78.9 | 70.8 / 78.9 |
| ReasoNet (Shen et al., 2017b) | 69.1 / 78.9 | 70.6 / 79.4 |
| Document Reader (Chen et al., 2017) | 70.0 / 79.0 | 70.7 / 79.4 |
| Ruminating Reader (Gong & Bowman, 2017) | 70.6 / 79.5 | 70.6 / 79.5 |
| jNet (Zhang et al., 2017) | 70.6 / 79.8 | 70.6 / 79.8 |
| Conductor-net | N/A | 72.6 / 81.4 |
| Interactive AoA Reader (Cui et al., 2017) | N/A | 73.6 / 81.9 |
| Reg-RaSoR | N/A | 75.8 / 83.3 |
| DCN+ | N/A | 74.9 / 82.8 |
| AIR-FusionNet | N/A | 76.0 / 83.9 |
| R-Net (Wang et al., 2017) | 72.3 / 80.7 | 76.5 /84.3 |
| BiDAF + Self Attention + ELMo | N/A | **77.9/ 85.3** |
| Reinforced Mnemonic Reader (Hu et al., 2017) | 73.2 / 81.8 | 73.2 / 81.8 |
| Dev set: QANet | **73.6 / 82.7** | N/A |
| Dev set: QANet + data augmentation ×2 | **74.5 / 83.2** | N/A |
| Dev set: QANet + data augmentation ×3 | **75.1 / 83.8** | N/A |
| Test set: QANet + data augmentation ×3 | **76.2 / 84.6** | 76.2 / 84.6 |

Table 2: The performances of different models on SQuAD dataset.

LSTMs as is used in most existing models. Specifically, each (embedding and model) encoder block is replaced with a 1, 2, or 3 layer Bidirectional LSTMs respectively, as such layer numbers fall into the usual range of the reading comprehension models (Chen et al., 2017). All of these LSTMs have hidden size 128. The results of the speedup comparison are shown in Table 3. We can easily see that our model is significantly faster than all the RNN based models and the speedups range from 3 to 13 times in training and 4 to 9 times in inference.

| | QANet | RNN-1-128 | Speedup | RNN-2-128 | Speedup | RNN-3-128 | Speedup |
|---|---|---|---|---|---|---|---|
| Training | **3.2** | 1.1 | **2.9x** | 0.34 | **9.4x** | 0.24 | **13.3x** |
| Inference | **8.1** | 2.2 | **3.7x** | 1.3 | **6.2x** | 0.92 | **8.8x** |

Table 3: Speed comparison between our model and RNN-based models on SQuAD dataset, all with batch size 32. RNN-$x$-$y$ indicates an RNN with $x$ layers each containing $y$ hidden units. Here, we use bidirectional LSTM as the RNN. The speed is measured by batches/second, so higher is faster.

**Speedup over BiDAF model.** In addition, we also use the same hardware (a NVIDIA p100 GPU) and compare the training time of getting the same performance between our model and the BiDAF model[14](Seo et al., 2016), a classic RNN-based model on SQuAD. We mostly adopt the default settings in the original code to get its best performance, where the batch sizes for training and inference are both 60. The only part we changed is the optimizer, where Adam with learning 0.001 is used here, as with Adadelta we got a bit worse performance. The result is shown in Table 4 which shows that our model is 4.3 and 7.0 times faster than BiDAF in training and inference speed. Besides, we only need one fifth of the training time to achieve BiDAF's best F1 score (77.0) on dev set.

---

[14]The code is directly downloaded from `https://github.com/allenai/bi-att-flow`

|        | Train time to get 77.0 F1 on Dev set | Train speed | Inference speed |
|--------|--------------------------------------|-------------|-----------------|
| QANet  | 3 hours                              | 102 samples/s | 259 samples/s |
| BiDAF  | 15 hours                             | 24 samples/s  | 37samples/s   |
| Speedup | **5.0x**                            | **4.3x**    | **7.0x**        |

Table 4: Speed comparison between our model and BiDAF (Seo et al., 2016) on SQuAD dataset.

### 4.1.3 ABALATION STUDY AND ANALYSIS

We conduct ablation studies on components of the proposed model, and investigate the effect of augmented data. The validation scores on the development set are shown in Table 5. As can be seen from the table, the use of convolutions in the encoders is crucial: both F1 and EM drop drastically by almost 3 percent if it is removed. Self-attention in the encoders is also a necessary component that contributes 1.4/1.3 gain of EM/F1 to the ultimate performance. We interpret these phenomena as follows: the convolutions capture the local structure of the context while the self-attention is able to model the global interactions between text. Hence they are complimentary to but cannot replace each other. The use of separable convolutions in lieu of tradition convolutions also has a prominent contribution to the performance, which can be seen by the slightly worse accuracy caused by replacing separable convolution with normal convolution.

**The Effect of Data Augmentation.** We additionally perform experiments to understand the values of augmented data as their amount increases. As the last block of rows in the table shows, data augmentation proves to be helpful in further boosting performance. Making the training data twice as large by adding the En-Fr-En data only (ratio 1:1 between original training data and augmented data, as indicated by row "data augmentation $\times$ 2 (1:1:0)") yields an increase in the F1 by 0.5 percent. While adding more augmented data with French as a pivot does not provide performance gain, injecting additional augmented data En-De-En of the same amount brings another 0.2 improvement in F1, as indicated in entry "data augmentation $\times$ 3 (1:1:1)". We may attribute this gain to the diversity of the new data, which is produced by the translator of the new language.

**The Effect of Sampling Scheme.** Although injecting more data beyond $\times$ 3 does not benefit the model, we observe that a good sampling ratio between the original and augmented data during training can further boost the model performance. In particular, when we increase the sampling weight of augmented data from (1:1:1) to (1:2:1), the EM/F1 performance drops by 0.5/0.3. We conjecture that it is due to the fact that augmented data is noisy because of the back-translation, so it should not be the dominant data of training. We confirm this point by increasing the ratio of the original data from (1:2:1) to (2:2:1), where 0.6/0.5 performance gain on EM/F1 is obtained. Then we fix the portion of the augmented data, and search the sample weight of the original data. Empirically, the ratio (3:1:1) yields the best performance, with 1.5/1.1 gain over the base model on EM/F1. This is also the model we submitted for test set evaluation.

### 4.1.4 ROBUSTNESS STUDY

In the following, we conduct experiments on the adversarial SQuAD dataset (Jia & Liang, 2017) to study the robustness of the proposed model. In this dataset, one or more sentences are appended to the original SQuAD context of test set, to intentionally mislead the trained models to produce wrong answers. However, the model is agnostic to those adversarial examples during training. We focus on two types of misleading sentences, namely, AddSent and AddOneSent. AddSent generates sentences that are similar to the question, but not contradictory to the correct answer, while AddOneSent adds a random human-approved sentence that is not necessarily related to the context.

The model in use is exactly the one trained with the original SQuAD data (the one getting 84.6 F1 on test set), but now it is submitted to the adversarial server for evaluation. The results are shown in Table 6, where the F1 scores of other models are all extracted from Jia & Liang (2017).[15] Again, we only compare the performance of single models. From Table 6, we can see that our model is on par with the state-of-the-art model Mnemonic, while significantly better than other models by a large margin. The robustness of our model is probably because it is trained with augmented data.

---

[15]Only F1 scores are reported in Jia & Liang (2017)

| | EM / F1 | Difference to Base Model EM / F1 |
|---|---|---|
| Base QANet | 73.6 / 82.7 | |
| - convolution in encoders | 70.8 / 80.0 | -2.8 / -2.7 |
| - self-attention in encoders | 72.2 / 81.4 | -1.4 / -1.3 |
| replace sep convolution with normal convolution | 72.9 / 82.0 | - 0.7 / -0.7 |
| + data augmentation ×2 (1:1:0) | 74.5 / 83.2 | +0.9 / +0.5 |
| + data augmentation ×3 (1:1:1) | 74.8 / 83.4 | +1.2 / +0.7 |
| + data augmentation ×3 (1:2:1) | 74.3 / 83.1 | +0.7 / +0.4 |
| + data augmentation ×3 (2:2:1) | 74.9 / 83.6 | +1.3 / +0.9 |
| + data augmentation ×3 (2:1:1) | 75.0 / 83.6 | +1.4 / +0.9 |
| + data augmentation ×3 (3:1:1) | **75.1 / 83.8** | **+1.5 / +1.1** |
| + data augmentation ×3 (4:1:1) | 75.0 / 83.6 | +1.4 / +0.9 |
| + data augmentation ×3 (5:1:1) | 74.9 / 83.5 | +1.3 / +0.8 |

Table 5: An ablation study of data augmentation and other aspects of our model. The reported results are obtained on the *development set*. For rows containing entry "data augmentation", "$\times N$" means the data is enhanced to $N$ times as large as the original size, while the ratio in the bracket indicates the sampling ratio among the original, English-French-English and English-German-English data during training.

The injected noise in the training data might not only improve the generalization of the model but also make it robust to the adversarial sentences.

| Single Model | AddSent | AddOneSent |
|---|---|---|
| Logistic (Rajpurkar et al., 2016) | 23.2 | 30.4 |
| Match (Wang & Jiang, 2016) | 27.3 | 39.0 |
| SEDT (Liu et al., 2017a) | 33.9 | 44.8 |
| DCR (Yu et al., 2016) | 37.8 | 45.1 |
| BiDAF (Seo et al., 2016) | 34.3 | 45.7 |
| jNet (Zhang et al., 2017) | 37.9 | 47.0 |
| Ruminating (Gong & Bowman, 2017) | 37.4 | 47.7 |
| RaSOR (Lee et al., 2016) | 39.5 | 49.5 |
| MPCM (Wang et al., 2016) | 40.3 | 50.0 |
| ReasoNet (Shen et al., 2017b) | 39.4 | 50.3 |
| Mnemonic (Hu et al., 2017) | **46.6** | **56.0** |
| QANet | 45.2 | 55.7 |

Table 6: The F1 scores on the adversarial SQuAD test set.

## 4.2 Experiments on TriviaQA

In this section, we test our model on another dataset TriviaQA (Joshi et al., 2017), which consists of 650K context-query-answer triples. There are 95K distinct question-answer pairs, which are authored by Trivia enthusiasts, with 6 evidence documents (context) per question on average, which are either crawled from Wikipedia or Web search. Compared to SQuAD, TriviaQA is more challenging in that: 1) its examples have much longer context (2895 tokens per context on average) and may contain several paragraphs, 2) it is much noisier than SQuAD due to the lack of human labeling, 3) it is possible that the context is not related to the answer at all, as it is crawled by key words.

In this paper, we focus on testing our model on the subset consisting of answers from Wikipedia. According to the previous work (Joshi et al., 2017; Hu et al., 2017; Pan et al., 2017), the same model would have similar performance on both Wikipedia and Web, but the latter is five time larger. To keep the training time manageable, we omit the experiment on Web data.

Due to the multi-paragraph nature of the context, researchers also find that simple hierarchical or multi-step reading tricks, such as first predicting which paragraph to read and then apply models like BiDAF to pinpoint the answer within that paragraph (Clark & Gardner, 2017), can significantly boost the performance on TriviaQA. However, in this paper, we focus on comparing with the single-paragraph reading baselines only. We believe that our model can be plugged into other

multi-paragraph reading methods to achieve the similar or better performance, but it is out of the scope of this paper.

The Wikipedia sub-dataset contains around 92K training and 11K development examples. The average context and question lengths are 495 and 15 respectively. In addition to the full development set, the authors of Joshi et al. (2017) also pick a verified subset that all the contexts inside can answer the associated questions. As the text could be long, we adopt the data processing similar to Hu et al. (2017); Joshi et al. (2017). In particular, for training and validation, we randomly select a window of length 256 and 400 encapsulating the answer respectively. All the remaining setting are the same as SQuAD experiment, except that the training steps are set to 120K.

**Accuracy.** The accuracy performance on the development set is shown in Table 7. Again, we can see that our model outperforms the baselines in terms of F1 and EM on Full development set, and is on par with the state-of-the-art on the Verified dev set.

|  | Full | Verified |
|---|---|---|
| Single Model | EM / F1 | EM / F1 |
| Random (Joshi et al., 2017) | 12.7 / 22.5 | 13.8 / 23.4 |
| Classifier (Joshi et al., 2017) | 23.4 / 27.7 | 23.6 / 27.9 |
| BiDAF (Seo et al., 2016) | 40.3 / 45.7 | 46.5 /52.8 |
| MEMEN (Pan et al., 2017) | 43.2/ 46.9 | 49.3 / 55.8 |
| M-Reader (Hu et al., 2017)[*] | 46.9/ 52.9[*] | 54.5/ 59.5[*] |
| QANet | **51.1 / 56.6** | 53.3/ 59.2 |

Table 7: The development set performances of different *single-paragraph* reading models on the Wikipedia domain of TriviaQA dataset. Note that [*] indicates the result on test set.

**Speedup over RNNs.** In addition to accuracy, we also benchmark the speed of our model against the RNN counterparts. As Table 8 shows, not surprisingly, our model has 3 to 11 times speedup in training and 3 to 9 times acceleration in inference, similar to the finding in SQuAD dataset.

|  | QANet | RNN-1-128 | Speedup | RNN-2-128 | Speedup | RNN-3-128 | Speedup |
|---|---|---|---|---|---|---|---|
| Training | **1.8** | 0.41 | **4.4x** | 0.20 | **9.0x** | 0.11 | **16.4x** |
| Inference | **3.2** | 0.89 | **3.6x** | 0.47 | **6.8x** | 0.26 | **12.3x** |

Table 8: Speed comparison between the proposed model and RNN-based models on TriviaQA Wikipedia dataset, all with batch size 32. RNN-$x$-$y$ indicates an RNN with $x$ layers each containing $y$ hidden units. The RNNs used here are bidirectional LSTM. The processing speed is measured by batches/second, so higher is faster.

## 5 RELATED WORK

Machine reading comprehension and automated question answering has become an important topic in the NLP domain. Their popularity can be attributed to an increase in publicly available annotated datasets, such as SQuAD (Rajpurkar et al., 2016), TriviaQA (Joshi et al., 2017), CNN/Daily News (Hermann et al., 2015), WikiReading (Hewlett et al., 2016), Children Book Test (Hill et al., 2015), etc. A great number of end-to-end neural network models have been proposed to tackle these challenges, including BiDAF (Seo et al., 2016), r-net (Wang et al., 2017), DCN (Xiong et al., 2016), ReasoNet (Shen et al., 2017b), Document Reader (Chen et al., 2017), Interactive AoA Reader (Cui et al., 2017) and Reinforced Mnemonic Reader (Hu et al., 2017).

Recurrent Neural Networks (RNNs) have featured predominatnly in Natural Language Processing in the past few years. The sequential nature of the text coincides with the design philosophy of RNNs, and hence their popularity. In fact, all the reading comprehension models mentioned above are based on RNNs. Despite being common, the sequential nature of RNN prevent parallel computation, as tokens must be fed into the RNN in order. Another drawback of RNNs is difficulty modeling long dependencies, although this is somewhat alleviated by the use of Gated Recurrent Unit (Chung et al.,

2014) or Long Short Term Memory architectures (Hochreiter & Schmidhuber, 1997). For simple tasks such as text classification, with reinforcement learning techniques, models (Yu et al., 2017) have been proposed to skip irrelevant tokens to both further address the long dependencies issue and speed up the procedure. However, it is not clear if such methods can handle complicated tasks such as Q&A. The reading comprehension task considered in this paper always needs to deal with long text, as the context paragraphs may be hundreds of words long. Recently, attempts have been made to replace the recurrent networks by full convolution or full attention architectures (Kim, 2014; Gehring et al., 2017; Vaswani et al., 2017b; Shen et al., 2017a). Those models have been shown to be not only faster than the RNN architectures, but also effective in other tasks, such as text classification, machine translation or sentiment analysis.

To the best of our knowledge, our paper is the first work to achieve both *fast* and *accurate* reading comprehension model, by discarding the recurrent networks in favor of feed forward architectures. Our paper is also the first to mix self-attention and convolutions, which proves to be empirically effective and achieves a significant gain of 2.7 F1. Note that Raiman & Miller (2017) recently proposed to accelerate reading comprehension by avoiding bi-directional attention and making computation conditional on the search beams. Nevertheless, their model is still based on the RNNs and the accuracy is not competitive, with an EM 68.4 and F1 76.2. Weissenborn et al. (2017) also tried to build a fast Q&A model by deleting the context-query attention module. However, it again relied on RNN and is thus intrinsically slower than ours. The elimination of attention further has sacrificed the performance (with EM 68.4 and F1 77.1).

Data augmentation has also been explored in natural language processing. For example, Zhang et al. (2015) proposed to enhance the dataset by replacing the words with their synonyms and showed its effectiveness in text classification. Raiman & Miller (2017) suggested using type swap to augment the SQuAD dataset, which essentially replaces the words in the original paragraph with others with the same type. While it was shown to improve the accuracy, the augmented data has the same syntactic structure as the original data, so they are not sufficiently diverse. Zhou et al. (2017) improved the diversity of the SQuAD data by generating more questions. However, as reported by Wang et al. (2017), their method did not help improve the performance. The data augmentation technique proposed in this paper is based on paraphrasing the sentences by translating the original text back and forth. The major benefit is that it can bring more syntactical diversity to the enhanced data.

## 6 CONCLUSION

In this paper, we propose a fast and accurate end-to-end model, QANet, for machine reading comprehension. Our core innovation is to completely remove the recurrent networks in the encoder. The resulting model is fully feedforward, composed entirely of separable convolutions, attention, linear layers, and layer normalization, which is suitable for parallel computation. The resulting model is both fast and accurate: It surpasses the best published results on SQuAD dataset while up to 13/9 times faster than a competitive recurrent models for a training/inference iteration. Additionally, we find that we are able to achieve significant gains by utilizing data augmentation consisting of translating context and passage pairs to and from another language as a way of paraphrasing the questions and contexts.

## ACKNOWLEDGEMENT

Adams Wei Yu is supported by NVIDIA PhD Fellowship and CMU Presidential Fellowship. We would like to thank Samy Bengio, Lei Huang, Minjoon Seo, Noam Shazeer, Ashish Vaswani, Barret Zoph and the Google Brain Team for helpful discussions.

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
