# OpenReview forum: "QANet: Combining Local Convolution with Global Self-Attention for Reading Comprehension"
_ICLR.cc/2018/Conference — Accept (Poster)_

### Official Review · AnonReviewer3 · 2017-11-25
**Good ideas, but not so good evaluation.**

**Rating:** 5
**Confidence:** 4

**Review:**

This paper proposes two contributions: first, applying CNNs+self-attention modules instead of LSTMs, which could result in significant speedup and good RC performance; second, enhancing the RC model training with passage paraphrases generated by a neural paraphrasing model, which could improve the RC performance marginally.

Firstly, I suggest the authors rewrite the end of the introduction. The current version tends to mix everything together and makes the misleading claim. When I read the paper, I thought the speeding up mechanism could give both speed up and performance boost, and lead to the 82.2 F1. But it turns out that the above improvements are achieved with at least three different ideas: (1) the CNN+self-attention module; (2) the entire model architecture design; and (3) the data augmentation method.

Secondly, none of the above three ideas are well evaluated in terms of both speedup and RC performance, and I will comment in details as follows:

(1) The CNN+self-attention was mainly borrowing the idea from (Vaswani et al., 2017a) from NMT to RC. The novelty is limited but it is a good idea to speed up the RC models. However, as the authors hoped to claim that this module could contribute to both speedup and RC performance, it will be necessary to show the RC performance of the same model architecture, but replacing the CNNs with LSTMs. Only if the proposed architecture still gives better results, the claims in the introduction can be considered correct.

(2) I feel that the model design is the main reason for the good overall RC performance. However, in the paper there is no motivation about why the architecture was designed like this. Moreover, the whole model architecture is only evaluated on the SQuAD dataset. As a result, it is not convincing that the system design has good generalization. If in (1) it is observed that using LSTMs in the model instead of CNNs could give on par or better results, it will be necessary to test the proposed model architecture on multiple datasets, as well as conducting more ablation tests about the model architecture itself.

(3) I like the idea of data augmentation with paraphrasing. Currently, the improvement is only marginal, but there seems many other things to play with. For example, training NMT models with larger parallel corpora; training NMT models with different language pairs with English as the pivot; and better strategies to select the generated passages for data augmentation.

I am looking forward to the test performance of this work on SQuAD.

---

> ### Author Response · Authors · 2018-01-02
> **Response to Reviewer 3**
>
> We believe there are misunderstandings we have addressed below. We also have also included more experimental results.
>
> Q: The reviewer said “I suggest the authors rewrite the end of the introduction. The current version tends to mix everything together and makes the misleading claim.”
> A: Thank you for the suggestions! We have revised the introduction to make our contributions clearer. Note that even though self-attention has already been used extensively in Vaswani et al, the combination of convolutions and self-attention is novel, and is significantly better than self-attention alone and gives 2.7 F1 gain in our experiments. The use of convolutions also allows us to take advantage of common regularization methods in ConvNets such as stochastic depth (layer dropout), which gives an additional gain of 0.2 F1 in our experiments.
> We would like to point out that in this paper the use of CNN + self-attention is to speed-up the model during training and inference.  The speed-up leads to faster experimentation and allows us to train on more augmented data, contributing to the strong result on SQuAD.
>
> Q: The reviewer comments “I feel that the model design is the main reason for the good overall RC performance. However, in the paper there is no motivation about why the architecture was designed like this.”
> A: At a high level, our architecture is the standard “embedding -> embedding encoder -> attention -> modeling encoder -> output” architecture, shared by many neural reading comprehension models. Thus, we do NOT claim any novelty in the overall architecture. Traditionally, the encoder components are bidirectional LSTMs. Our motivation is to speed-up the architecture by replacing the bidirectional LSTMs with convolution+self-attention for the encoders for both embedding and modeling components. The context passages are over one hundred words long in SQuAD, so the parallel nature of CNN architectures leads to a significant speed boost for both training and inference. Replacing bidirectional LSTMs with convolution+self-attention is our main novelty.
>
> Q: The reviewer comments “it will be necessary to show the RC performance of the same model architecture, but replacing the CNNs with LSTMs. Only if the proposed architecture still gives better results, the claims in the introduction can be considered correct.”
> A: We think the reviewer might have misunderstood our claim. As mentioned above, we do NOT claim any novelty in the overall architecture, as it is a common reading comprehension model. We will make this point clearer in the revision. Our contribution, as we have emphasized several times, is to replace the LSTM encoders with convolution+self-attention, without changing the remaining components. We find the resulting model both fast and accurate. In fact, if we switch back to LSTM encoders, it will become BiDAF [1] or DCN [2], which are both slower (see our speedup experiments) and less accurate (see the leaderboard: https://rajpurkar.github.io/SQuAD-explorer/) than ours.
>
> [1] Bidirectional Attention Flow for Machine Comprehension. In ICLR 2017.
> Minjoon Seo, Aniruddha Kembhavi, Ali Farhadi, Hannaneh Hajishirzi.
> [2] Dynamic Coattention Networks For Question Answering. ICLR 2017.
> Caiming Xiong, Victor Zhong, Richard Socher.
>
> Q: Results on one more dataset.
> A: We have conducted experiments on another Q&A dataset, TriviaQA, to verify that the effectiveness and efficiency of our model is general. In a nutshell, again, our model is 4x to 16x times faster than the RNN counterparts, while outperforming the state-of-the-art single-paragraph-reading model by more than 3.0 in both F1 and EM. Please see the revision.
>
> Q: More result on data augmentation.
> A: Thanks for the suggestions! We indeed put more experiments in the revision and here are some interesting findings:
> Translating to more languages can lead to more diverse augmented data, which further result in better generalization. Currently we try both French and German.
> The sampling ratio of (original : English-French-English : English-German-English) during training matters. The best empirical ratio is 3:1:1.
>
> Q: Leaderboard result.
> A: We submitted our best model for test set evaluation on SQuAD, on Dec 20, 2017. Our single model (named “FRC”) is ranked 3rd among all single models in terms of F1 with F1/EM=84.6/76.2 (https://rajpurkar.github.io/SQuAD-explorer/). The performance gain is because we add more regularization to the model. Note that the two single models ranked above us have NOT been published yet: “BiDAF + Self Attention + ELMo” & “AttentionReader+”.

---

### Official Review · AnonReviewer2 · 2017-11-27
**Interesting augmentation method**

**Rating:** 6
**Confidence:** 3

**Review:**

This paper presents a reading comprehension model using convolutions and attention. This model does not use any recurrent operation but it is not per se simpler than a recurrent model. Furthermore, the authors proposed an interesting idea to augment additional training data by paraphrasing based on off-the-shelf neural machine translation.  On SQuAD dataset, their results show some small improvements using the proposed augmentation technique. Their best results, however, do not outperform the best results reported on the leader board.

Overall, this is an interesting study on SQuAD dataset. I would like to see results on more datasets and more discussion on the data augmentation technique. At the moment, the description in section 3 is fuzzy in my opinion. Interesting information could be:
- how is the performance of the NMT system?
- how many new data points are finally added into the training data set?
- what do ‘data aug’ x 2 or x 3 exactly mean?

---

> ### Author Response · Authors · 2018-01-02
> **Response to Reviewer 2**
>
> From the reviewer’s comments, it is not immediately clear to us the reviewer’s rationale for rejection. What we only know is the reviewer wants to know more about the data augmentation approach. It would be great if the reviewer can elaborate more on the rejection rationale.
>
> We believe our work is significant in the following aspects:
> (a) Our work is novel: we introduced a new architecture for reading comprehension and a data augmentation technique that yields non-trivial gain on a strong SQuAD model. Note that even though self-attention has already been used extensively in Vaswani et al, the combination of convolutions and self-attention is novel, and is significantly better than self-attention alone and gives 2.7 F1 gain in our experiments. The use of convolutions also allows us to take advantage of common regularization methods in ConvNets such as stochastic depth (layer dropout), which gives an additional gain of 0.2 F1 in our experiments.
>
> (b) Our model is accurate: we are currently ranked 3rd by F1 score on the SQuAD leaderboard among single models (note: the two single models ranked above us are not published yet).
>
> (c) Our model is fast: we achieve a speed-up of up to 13x and 9x in training and inference respectively on SQuAD.
>
> As stated above, we are disappointed with the low scores that our paper has received. Concurrent to our submission, there are two other papers on SQuAD, FusionNet[1] and DCN+ [2], which only tested on SQuAD and obtained much lower F1 scores (83.9 and 83.0 respectively) compared to ours (84.6). Their papers, however, received the averaged review scores of 6.33 and 7 respectively, which are much higher than our averaged review score of 5.33. As such, we encourage the reviewers to reconsider their scores.
>
> [1] https://openreview.net/forum?id=BJIgi_eCZ&noteId=BJIgi_eCZ
> [2] https://openreview.net/forum?id=H1meywxRW
>
> More detailed comments:
> Q: Regarding “simplicity”.
> A: Thanks for raising this point. By simplicity, we mean we do not use hand-crafted features such as POS tagging ([3]), nor multiple reading pass ([4]). We have made this point clear in the revision and tried not using “simple” to avoid confusion.
>
> [3] Reading Wikipedia to Answer Open-Domain Questions. In ACL 2017.
> Danqi Chen, Adam Fisch, Jason Weston, Antoine Bordes.
> [4] Reasonet: Learning to stop reading in machine comprehension. In KDD 2017.
> Yelong Shen, Po-Sen Huang, Jianfeng Gao, Weizhu Chen.
>
> Q: Leaderboard result.
> A: We submitted our best model for test set evaluation on SQuAD, on Dec 20, 2017. Our single model (named “FRC”) is ranked 3rd among all single models in terms of F1 with F1/EM=84.6/76.2 (https://rajpurkar.github.io/SQuAD-explorer/). The performance gain is because we add more regularization to the model. Note that the two single models ranked above us have NOT been published yet: “BiDAF + Self Attention + ELMo” & “AttentionReader+”.
>
> Q: Results on one more dataset.
> A: We have conducted experiments on another Q&A dataset, TriviaQA, to verify that the effectiveness and efficiency of our model is general. In a nutshell, again, our model is 4x to 16x times faster than the RNN counterparts, while outperforming the state-of-the-art single-paragraph-reading model by more than 3.0 in both F1 and EM. Please see the revision.
>
> Q: Section 3 and more discussion on the data augmentation.
> A: We have revised the paper to give more details regarding our method and results with data augmentation. Here, we highlight a few major details that the reviewers asked, as well as several new findings:
> a) Performance of NMT systems:
> English-German (newstest2015): 27.6 (to German) and 29.9 (to English)
> English-French (newstest2014): 36.7 (to French) and 35.9 (to English)
> b) Note that in our Table, “x2” means the total amount of the final training data is twice as large as the original data, i.e. the added amount is the same as the original. We have clarified this as well in the revision.
> c) New finding: translating to more languages can lead to more diverse augmented data, which further result in better generalization. Currently we try both English-French and English-German.
> d) New finding: we have shown in the revised experiment section that different ratios (original : English-French-English : English-German-English) would have different effects on the final performance. Empirically,  when the ratio is 3:1:1, we get the best result. We interpret this phenomenon as: the translation might bring noise to the augmented data that we should lay more weight to the original clean data.

---

### Official Review · AnonReviewer1 · 2017-11-27
**Good paper. Results in an additional dataset needed.**

**Rating:** 8
**Confidence:** 5

**Review:**

Summary:

This paper proposes a non-recurrent model for reading comprehension which used only convolutions and attention. The goal is to avoid recurrent which is sequential and hence a bottleneck during both training and inference. Authors also propose a paraphrasing based data augmentation method which helps in improving the performance. Proposed method performs better than existing models in SQuAD dataset while being much faster in training and inference.

My Comments:

The proposed model is convincing and the paper is well written.

1. Why don’t you report your model performance without data augmentation in Table 1? Is it because it does not achieve SOTA? The proposed data augmentation is a general one and it can be used to improve the performance of other models as well. So it does not make sense to compare your model + data augmentation against other models without data augmentation. I think it is ok to have some deterioration in the performance as you have a good speedup when compared to other models.

2. Can you mention your leaderboard test accuracy in the rebuttal?

3. The paper can be significantly strengthened by adding at least one more reading comprehension dataset. That will show the generality of the proposed architecture. Given the sufficient time for rebuttal, I am willing to increase my score if authors report results in an additional dataset in the revision.

4. Are you willing to release your code to reproduce the results?


Minor comments:

1. You mention 4X to 9X for inference speedup in abstract and then 4X to 10X speedup in Intro. Please be consistent.
2. In the first contribution bullet point, “that exclusive built upon” should be “that is exclusively built upon”.

---

> ### Author Response · Authors · 2018-01-02
> **Response to Reviewer 1**
>
> We thank the reviewer for your acknowledgement to our contributions! We address the comments below.
>
> Q: The reviewer asks “Why don’t you report your model performance without data augmentation in Table 1?”
> A: We thank the reviewer for the suggestion! We have added this result in the revision. In summary, without data augmentation, our model gets 82.7 F1 on dev set, while with data augmentation, we get 83.8 F1 on dev. We only submitted the model with augmented data, and get 84.6 F1 on test set, which outperforms most of the existing models and is the best among all the published results, as of Dec 20, 2017.
>
> Q: The reviewer asks “Can you mention your leaderboard test accuracy in the rebuttal?”
> A: We submitted our best model for test set evaluation on SQuAD, on Dec 20, 2017. Our single model (named “FRC”) is ranked 3rd among all single models in terms of F1 with F1/EM=84.6/76.2 (https://rajpurkar.github.io/SQuAD-explorer/). The performance gain is because we add more regularization to the model. Note that the two single models ranked above us have NOT been published yet: “BiDAF + Self Attention + ELMo” & “AttentionReader+”.
>
> Q: Results on one more dataset.
> A: We have conducted experiments on another Q&A dataset, TriviaQA, to verify that the effectiveness and efficiency of our model is general. In a nutshell, again, our model is 4x to 16x times faster than the RNN counterparts, while outperforming the state-of-the-art single-paragraph-reading model by more than 3.0 in both F1 and EM. Please see the revision.
>
> Q: The reviewer asks “Are you willing to release your code to reproduce the results?”
> A: Yes, we will release the code after the paper gets accepted.
>
> Q: Minor comments.
> A: Thank you. We addressed all of them in the latest revision.

---

> > ### Author Response · Authors · 2018-01-03
> > **Additional Dataset**
> >
> > As we have included the result on the triviaQA dataset as well, we hope the reviewer can reconsider the score, as promised in the original review. Thanks again for your suggestion to help us improve the paper!

---

> > > ### Comment · AnonReviewer1 · 2018-01-12
> > > **happy with rebuttal**
> > >
> > > I am happy with the rebuttal. I think this paper has good enough contributions to get published.
> > >
> > > I have revised my scores accordingly.

---

### Public Comment · (anonymous) · 2017-11-14
**Implementation details**

Thank you for your work. It seems the paper lacks some of the implementation details and sometimes includes ambiguous statements.
1. What is the number of heads used for the multi-head self attention, and is the number consistent throughout the layers? And is the attention key depth per head also 128? I feel that the encoder layer detail is lacking.
2. Subsection 2.2 in 2. Embedding Encoder Layer, the paper states that kernel size of 7 is used for embedding encoder. However, later on subsection 4.2 Basic Setup describes "the kernel sizes are 5 and 7" respectively. Could you please clarify this?

---

> ### Author Response · Authors · 2017-11-14
> **Response to Implementation details**
>
> Thanks for your interest and the questions! Here are the answers:
>
> 1.  The number of heads is 8, which is consistent throughout the layers. The attention key depth is 128, so the per head depth is 128/8=16.
>
> 2. It should be "the kernel sizes are 7 and 5".
>
> We will clarify those in the revision. Thanks!

---

### Comment · Area_Chair · 2017-12-28
**Rebuttal**

Authors,

Please post a rebuttal for this work. Discussion period ends Jan 5th.

---

> ### Author Response · Authors · 2018-01-02
> **We submitted the rebuttal and revision**
>
> Dear Area Chair,
>
> We have submitted the rebuttal and revision. Our rebuttal contains a general one to address the common concerns of the reviewers, and three separated ones to answer the individual questions for each reviewer.
>
> Thanks!

---

### Author Response · Authors · 2018-01-02
**General comments (novelty, leaderboard, additional benchmark):**

We thank reviewers for comments and feedback to our paper, which have helped us improve the paper. However, we are disappointed with the low scores that our paper has received. Concurrent to our submission, there are two other papers on SQuAD, FusionNet[1] and DCN+ [2], which only tested on SQuAD and obtained much lower F1 scores (83.9 and 83.0 respectively) compared to ours (84.6). Their papers, however, received the averaged review scores of 6.33 and 7 respectively, which are much higher than our averaged review score of 5.33. As such, we encourage the reviewers to reconsider their scores.

[1] https://openreview.net/forum?id=BJIgi_eCZ&noteId=BJIgi_eCZ
[2] https://openreview.net/forum?id=H1meywxRW

We answer here some key questions by the reviewers:

1. Novelty
A major concern amongst the reviewers novelty of this paper because it’s similar to Vaswani et al. We stress here that our model is indeed novel: Note that even though self-attention has already been used extensively in Vaswani et al, the combination of convolutions and self-attention is novel, and is significantly better than self-attention alone and gives 2.7 F1 gain in our experiments. Our good accuracy is coupled with very good speedup gains. The speedup gains of up to 13x per training iteration and 9x during inference on SQuAD is not small. This significant gain makes our model most promising for larger datasets.

2. Test set result on SQuAD leaderboard
We submitted our best model for test set evaluation on SQuAD, on Dec 20, 2017. Our single model (named “FRC”) is ranked 3rd among all single models in terms of F1 with F1/EM=84.6/76.2 (https://rajpurkar.github.io/SQuAD-explorer/). The performance gain is because we add more regularization to the model. Note that the two single models ranked above us have NOT been published yet: “BiDAF + Self Attention + ELMo” & “AttentionReader+”.

3. Results on an additional benchmark (TriviaQA)
We have conducted experiments on another Q&A dataset, TriviaQA, to verify that the effectiveness and efficiency of our model is general. In a nutshell, again, our model is 4x to 16x times faster than the RNN counterparts, while outperforming the state-of-the-art single-paragraph-reading model by more than 3.0 in both F1 and EM. Please see the revision.

---

### Author Response · Authors · 2018-01-06
**General comments 2 (Result on Adversarial SQuAD dataset added)**

We have just added a new result on the adversarial SQuAD dataset [1] . In terms of robustness to the adversarial examples, our model is on par with the state-of-the-art model. Please see the Section 4.1.5 of the latest version for more details.

ps: This addition is also partly motivated by Reviewer 1's promise to increase our score (to above 7). Until now, we have added 2 more benchmarks: TriviaQA & Adversarial SQuAD.

[1] Jia Robin, Percy Liang. Adversarial Examples for Evaluating Reading Comprehension Systems. In EMNLP 2017.

Thanks!

---

### Public Comment · (anonymous) · 2018-01-20
**Training vs dev ratio**

The performance of the model on SQuAD dataset is impressive. In addition to the performance on the test set, we are also interested in the sample complexity of the proposed model. Currently, the SQuAD dataset splits the collection of passages into a training set, a development set, and a test set in a ratio of 80%:10%:10% where the test set is not released. Given the released training and dev set, we are wondering what would happen if we split the data in a different ratio, for example, 50% for training and the rest 50% for dev. We will really appreciate it if the authors can report the model performance (on training/dev respectively) under this scenario.

---

### Public Comment · (anonymous) · 2018-01-24
**Speed-up comparison against BiDAF**

Thank you for your paper! We really liked your approach to accelerate inference and training times in QA.
I have one question regarding the comparison with BiDAF. On the article, you mention that you batched the training examples by paragraph length in your model, but it is not clear whether you did the same for BiDAF (the implementation on GitHub offers the flags --cluster and --len_opt for that).
That is an important consideration because that change alone has a significant impact on training and inference times. In fact, by batching the inputs in that way we have fully trained BiDAF in 6h30 (12 epochs) on an NVidia Titan X, which is more than two times faster than your reported time of 15h on a more powerful P100.
Could you please clarify this point in the article?

---

> ### Author Response · Authors · 2018-02-24
> **Reply**
>
> Hi,
>
> Thanks for your question!
>
> We did set the flag --cluster and len_opt to be true. The potential explanation for the difference might be:
>
> 1. We used the Adam optimizer with learning rate 0.001 instead of the Adadelta optimizer with learning rate 0.5, as we found the Adadelta can lead to a bit worse performance.
>
> 2. The dev set score 77.0 is obtained from the evaluation of BiDAF, not from the official SQuAD evaluation script. In a personal communication with Minjoon Seo, the first author of BiDAF in June 2017, he mentioned the BiDAF evaluation is a bit harsher than the official one. That may imply we need to train longer than what is reported in the paper to reach this accuracy. Besides, in our experiment, we found it took much longer to get 77.0 than 76.5.
>
> 3. In fact, if our goal is merely to achieve 77.0 (which is a relatively low accuracy) as fast as possible,  we can speed up our model significantly by making the encoders shallower. But it would lower our final accuracy. Therefore, maybe a more meaningful speed comparison is the 3rd and 4th column of table 4, as they are measured by the samples processed per second.
>
> Thanks!

---

### Decision · Program_Chairs · 2018-01-29
**ICLR 2018 Conference Acceptance Decision**

**Decision:**

Accept (Poster)

**Comment:**

This work replaces the RNN layer of square with a self-attention and convolution, achieving a big speed up and performance gains, particularly with data augmentation. The work is mostly clear presented, one reviewer found it "well-written" although there was a complaint the work did not clear separate out the novel aspects. In terms of results the work is clearly of high quality, producing top numbers on the shared task. There were some initial complaints of only using the SQuAD dataset, but the authors have now included additional results that diversify the experiments. Perhaps the largest concern is novelty. The idea of non-RNN self-attention is now widely known, and there are several systems that are applying it. Reviewers felt that while this system does it well, it is maybe less novel or significant than other possible work.